# Discovery of conjoined charge density waves in the kagome superconductor CsV₃Sb₅

Haoxiang Li [1,13] ✉, G. Fabbris [2], A. H. Said[2], J. P. Sun [3,4], Yu-Xiao Jiang[5], J.-X. Yin [6] ✉, Yun-Yi Pai [1], Sangmoon Yoon[1,14], Andrew R. Lupini [7], C. S. Nelson [8], Q. W. Yin[9], C. S. Gong[9], Z. J. Tu[9], H. C. Lei [9], J.-G. Cheng [3,4], M. Z. Hasan [5], Ziqiang Wang [10], Binghai Yan [11], R. Thomale [12], H. N. Lee [1] & H. Miao [1] ✉

The electronic instabilities in CsV₃Sb₅ are believed to originate from the V 3d-electrons on the kagome plane, however the role of Sb 5p-electrons for 3-dimensional orders is largely unexplored. Here, using resonant tender X-ray scattering and high-pressure X-ray scattering, we report a rare realization of conjoined charge density waves (CDWs) in CsV₃Sb₅, where a $2 \times 2 \times 1$ CDW in the kagome sublattice and a Sb 5p-electron assisted $2 \times 2 \times 2$ CDW coexist. At ambient pressure, we discover a resonant enhancement on Sb $L_1$-edge ($2s \rightarrow 5p$) at the $2 \times 2 \times 2$ CDW wavevectors. The resonance, however, is absent at the $2 \times 2 \times 1$ CDW wavevectors. Applying hydrostatic pressure, CDW transition temperatures are separated, where the $2 \times 2 \times 2$ CDW emerges 4 K above the $2 \times 2 \times 1$ CDW at 1 GPa. These observations demonstrate that symmetry-breaking phases in CsV₃Sb₅ go beyond the minimal framework of kagome electronic bands near van Hove filling.

Charge density waves (CDWs), a translational symmetry-breaking electronic fluid state, are in the spotlight to unravel intertwined quantum materials. This includes cuprate high-$T_C$ superconductors[1–3], Moiré superlattices[4], and nonmagnetic kagome metals where the CDW emerges as a leading electronic instability near van Hove filling[5–12]. Intriguingly, due to the sublattice interference on the geometrically frustrated triangle network, CDWs are predicted to potentially feature finite angular momentum or time-reversal symmetry breaking[7–12]. Recently, an exotic CDW in combination with superconductivity (SC) has been discovered in a kagome superconductor family $AV_3Sb_5$ ($A$ = K, Rb, Cs) (Fig. 1a)[13–25]. At ambient pressure, a CDW in $AV_3Sb_5$ sets in between 78 K and 103 K. Electronic nematicity and time-reversal symmetry breaking that are potentially related to the CDW phase have been observed[15,16,22,24,25]. Below $T_{SC}$~3 K, SC emerges and displays an unconventional pair-density modulation in real space[23].

While it is apparent that the spatial symmetry breaking plays a key role in describing the exotic electronic phases in CsV₃Sb₅, the origin of CDW and its relationship with electronic nematicity and SC are yet to

[1]Materials Science and Technology Division, Oak Ridge National Laboratory, Oak Ridge, TN 37831, USA. [2]Advanced Photon Source, Argonne National Laboratory, Argonne, IL 60439, USA. [3]Beijing National Laboratory for Condensed Matter Physics and Institute of Physics, Chinese Academy of Sciences, Beijing 100190, China. [4]School of Physical Sciences, University of Chinese Academy of Sciences, Beijing 100190, China. [5]Laboratory for Topological Quantum Matter and Advanced Spectroscopy (B7), Department of Physics, Princeton University, Princeton, NJ 08544, USA. [6]Laboratory for Quantum Emergence, Department of Physics, Southern University of Science and Technology, Shenzhen, Guangdong 518055, China. [7]Center for Nanophase Materials Sciences, Oak Ridge National Laboratory, Oak Ridge, TN 37831, USA. [8]National Synchrotron Light Source II, Brookhaven National Laboratory, Upton, NY 11973, USA. [9]Department of Physics and Beijing Key Laboratory of Opto-Electronic Functional Materials and Microdevices, Renmin University of China, Beijing 100872, China. [10]Department of Physics, Boston College, Chestnut Hill, MA 02467, USA. [11]Department of Condensed Matter Physics, Weizmann Institute of Science, Rehovot 7610001, Israel. [12]Institute for Theoretical Physics, University of Würzburg, Am Hubland, D-97074 Würzburg, Germany. [13]Present address: Advanced Materials Thrust, The Hong Kong University of Science and Technology (Guangzhou), Guangzhou, Guangdong 511453, China. [14]Present address: Department of Physics, Gachon University, Seongnam 13120, Republic of Korea. ✉e-mail: haoxiangli@ust.hk; yinjx@sustech.edu.cn; miaoh@ornl.gov

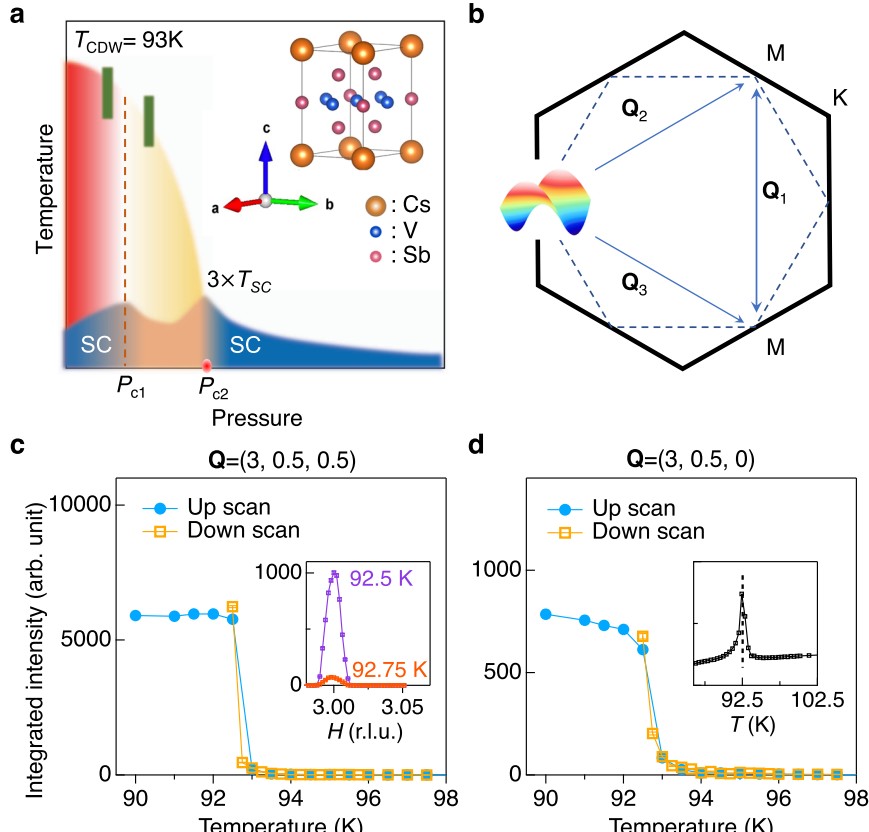

**Fig. 1 | Charge density wave in CsV₃Sb₅.** **a** $T$-$P$ phase diagram and crystal structure of CsV₃Sb₅ (space group P6 = mmm, no. 191). The phase diagram is separated by two characteristic pressures, $P_{c1}$ and $P_{c2}$. The curvature of resistivity at $T_{CDW}$ changes sign at $P_{c1}$. At the same pressure, superconductivity reaches its first peak. The green markers indicate pressures where the high-pressure X-ray diffraction measurement was performed (Fig. 3). **b** A schematic of van Hove singularities at the M point of the hexagonal Brillouin zone. Theoretically, the van Hove filling induce Fermi surface instabilities with three nesting wavevector $\mathbf{Q}_{1,2,3}$. Integrated CDW intensity vs temperature measured at $\mathbf{Q}_{CDW} = (3, 0.5\ 0.5)$ (**c**) and (3, 0.5, 0) (**d**) using meV-resolution elastic X-ray scattering. A sharp jump of the order parameter at the CDW transition indicates a weak first order phase transition and is consistent with nuclear magnetic resonance measurement[41]. The intensity jump is much stronger in the out-of-plane CDW peak at (3, 0.5, 0.5) compared to the in-plane one at (3.5, 0.5, 0). The inset of **c** shows the CDW peak intensity measured at 92.5 K and 92.75 K. The specific heat data shown in the inset of **d** reveal a sharp transition at 92.5 K, consistent with the diffraction data. The error bars in **c**, **d** represent 1-standard deviation assuming Poisson counting statistics.

be discovered. Like the conventional vs. unconventional paradigm for superconducting pairing, a CDW can be mediated by phonons or electronic interactions, which is challenging to discriminate. This particularly applies to electronic kagome bands, where the $3d$-van Hove singularities at the M point imply particle-hole fluctuations favoring a $2 \times 2 \times 1$ CDW that preserves the $C_6$ rotational symmetry (Fig. 1b)[5-9]. In experimental studies, the CDW in kagome metals appears to be 3-dimensional (3D) with a $2 \times 2 \times 2$ superstructure[17,18], indicating a non-trivial interlayer coupling that possibly breaks $C_6$ to $C_2$[11,12,26]. Still, a Landau theory analysis shows that the $2 \times 2 \times 2$ CDW alone is incompatible with the first order phase transition (Fig. 1c, d), which could point to a novel coexistence of $2 \times 2 \times 1$ and $2 \times 2 \times 2$ CDWs[11,12]. Turning to the superconducting phase, the intimate correlation between CDW and SC extends to the temperature ($T$) vs pressure ($P$) phase diagram[19-21]. As depicted in Fig. 1a, the CDW transition temperature, $T_{CDW}$, monotonically decreases and eventually vanishes at $P_{c2}$. Near an intermediate pressure $P_{c1}$-0.7 GPa, the curvature of the resistivity at $T_{CDW}$ changes sign. Interestingly, SC also shows two turning points at $P_{c1}$ and $P_{c2}$ and forms a double-dome structure. In this letter, we use advanced X-ray scattering to demonstrate a rare realization of conjoined CDWs in CsV₃Sb₅. The conjoined CDWs transcend the phenomenology that could be derived from a sole V kagome sublattice near van Hove filling and provides spatial symmetry constraints to understand the enlarged complexity of novel symmetry-breaking phases in CsV₃Sb₅.

## Results

We start by examining the electronic origin of this CDW using resonant elastic X-ray scattering (REXS). As depicted in Fig. 2a, by tuning the photon energy to atomic absorption edges, orbital-resolved valence electrons that are involved in the CDW will be enhanced[3]. For instance, in the cuprate high-$T_c$ superconductors, CDW intensity resonates at both O K edge ($1s{\to}2p$)[27] and Cu $L_3$ edge ($2p{\to}3d$)[28,29], reflecting the $d$-$p$ hybridized wave function. In CsV₃Sb₅, while the V $L_3$ edge ($h\nu = 0.512$ keV) is ideal to probe the $3d$-electron contributions for the CDW, the low-photon energy prevents its access to the CDW wavevectors. Instead, we choose the Sb $L_1$ edge ($2s{\to}5p$, $h\nu = 4.7$ keV) to enhance the $5p$-electrons near the Fermi level (see Supplementary Fig. 2 for REXS at other edges). At the Sb $L_1$ edge, both $\mathbf{Q}_{CDW}^{2\times2\times1}$ with $L = integer$ and $\mathbf{Q}_{CDW}^{2\times2\times2}$ with $L = half$-$integer$ can be reached[18]. Moreover, since Sb $5p_z$-electrons form a large Fermi surface and directly contribute to the coupling between adjacent kagome layers, the Sb $L_1$-edge resonance can potentially be used to distinguish the $2 \times 2 \times 1$ CDW in the kagome sublattice and the interlayer coupled $2 \times 2 \times 2$ CDW.

Our main observations at the Sb $L_1$ edge are summarized in Fig. 2b–e. All REXS data were collected in the reflection geometry at $T = 10$ K (see Supplementary Fig. 1). The Sb $L_1$ edge is identified around $h\nu = 4.7$ keV in the fluorescence scan shown in Fig. 2b. Figure 2c and d shows energy scans at fixed $\mathbf{Q}_{CDW}^{2\times2\times1} = (\pm 0.5, 0, 2)$ and $\mathbf{Q}_{CDW}^{2\times2\times2} = (\pm 0.5, 0, 2.5)$ (see data at more Q points in Fig. S1). Interestingly, we find a strong dip in scans at $2 \times 2 \times 1$ wavevectors, which is in

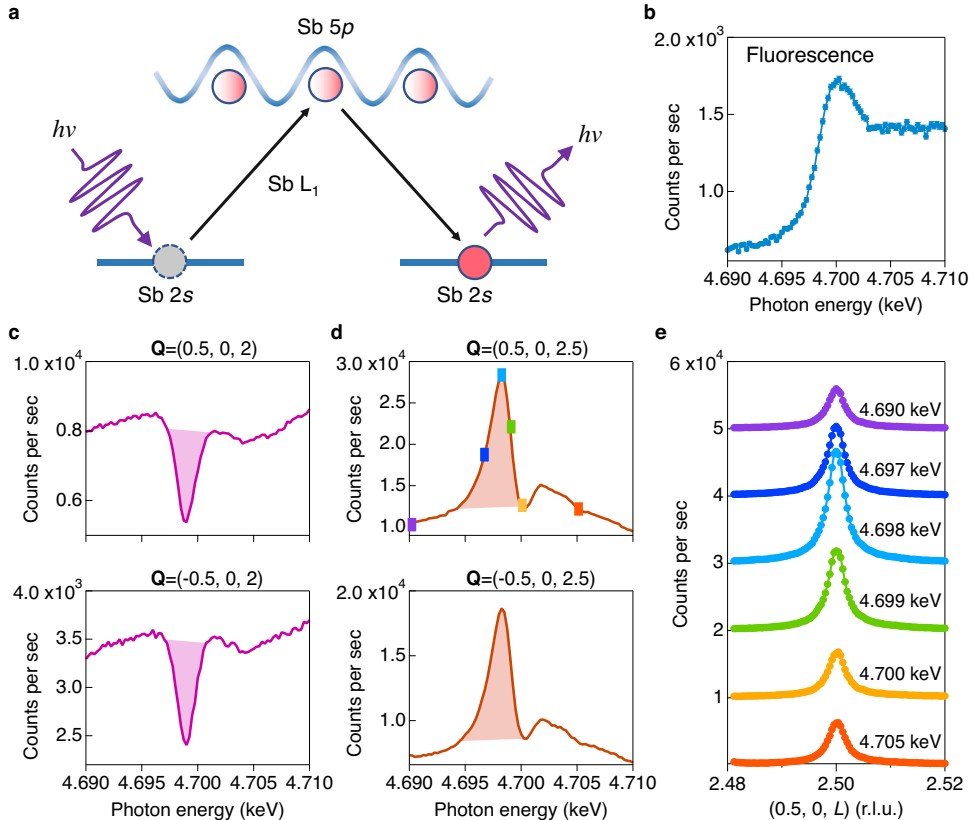

**Fig. 2 | Sb $L_1$-edge REXS reveals conjoined CDWs in CsV$_3$Sb$_5$. a** Schematic of the REXS process. The Sb 2s to 5p transition is allowed by the dipole selection rule. **b** X-ray fluorescence near the Sb $L_1$-edge (4.7 keV). **c, d** Photon energy scans at the $\mathbf{Q}_{\mathrm{CDW}}^{2\times2\times1}=(\pm0.5,0,2)$ and $\mathbf{Q}_{\mathrm{CDW}}^{2\times2\times2}=(\pm0.5,0,2.5)$ taken at $T=10$ K. Resonant peaks in the energy scan of the $2\times2\times2$ CDW demonstrate a Sb 5p-electron assisted CDW. This resonance can be directly observed in energy-dependent L-scans shown in **e**. The error bars in **b** represent 1-standard deviation assuming Poisson counting statistics.

stark contrast to scans at $2\times2\times2$ wavevectors that show large resonant enhancement slightly below the Sb $L_1$ edge. The dramatically different resonant response between $\mathbf{Q}_{\mathrm{CDW}}^{2\times2\times1}$ and $\mathbf{Q}_{\mathrm{CDW}}^{2\times2\times2}$ demonstrates different electronic origins of the $2\times2\times1$ and $2\times2\times2$ CDWs in CsV$_3$Sb$_5$, where the Sb 5p valence electrons are only involved in the formation of the $2\times2\times2$ CDW. An immediate consequence of two CDW order parameters is that the order parameter interference allows cubic free-energy terms in the Landau theory analysis[11,12] and hence naturally explains the puzzling weak first order CDW transition in CsV$_3$Sb$_5$.

Having established the two conjoined CDWs with different charge contributions, we examine the CDW evolution under pressure. The experimental geometry of high-pressure diffraction is depicted in Fig. 3a. Figure 3b shows the integrated CDW intensity at $P=0.5$ GPa $<P_{c1}$ (the green rectangles in Fig. 1a mark the location of the measurement in the $T$–$P$ phase diagram). Consistent with the ambient pressure measurement, the CDW peaks at $\mathbf{Q}_{\mathrm{CDW}}^{2\times2\times2}$ and $\mathbf{Q}_{\mathrm{CDW}}^{2\times2\times1}$ emerge at the same temperature around 77 K. Upon increasing pressure up to $P=1$ GPa $>P_{c1}$, however, the degeneracy of the conjoined CDWs is lifted with a 4 K gap between $T_{\mathrm{CDW}}^{2\times2\times2}$ and $T_{\mathrm{CDW}}^{2\times2\times1}$ (Fig. 3c). Figure 3d and e compares the $2\times2\times1$ and $2\times2\times2$ CDW intensities at 57 K and 61 K, respectively. The absence of the $2\times2\times1$ CDW peak at 61 K provides independent evidence for two CDW order parameters in CsV$_3$Sb$_5$. Unexpectedly, the $2\times2\times2$ CDW peak precedes the $2\times2\times1$ peak at $P>P_{c1}$, casting doubt on scenarios based on a simple stacking of $2\times2\times1$ superlattices with a π-phase shift, as this should lead to a higher $T_{\mathrm{CDW}}^{2\times2\times1}$ compared to $T_{\mathrm{CDW}}^{2\times2\times2}$. This observation further supports two CDW order parameters in CsV$_3$Sb$_5$ with different electronic origins. Indeed, our observation is consistent with a Landau theory analysis that showed two CDW transitions in the presence of order

parameter couplings[12]. The separation of the conjoined CDW coincides with the superconducting crossover region in the $T$ vs $P$ phase diagram shown in Fig. 1a (orange shaded area). This observation suggests that the $2\times2\times1$ CDW in the kagome sublattice is more closely related to the first superconducting dome, whereas the 5p-assisted $2\times2\times2$ CDW has more impact on the second superconducting dome.

The observation of conjoined CDWs in CsV$_3$Sb$_5$ may also resemble the intertwined cuprate high-$T_c$ superconductors, where under magnetic field[30,31] or unidirectional pressure[32] a short-ranged CDW broadly peak at $L=$ *half-integer* coexists with a long-ranged CDW at $L=$ *integer*. It should be noted, however, that the two conjoined CDWs in CsV$_3$Sb$_5$ are both long-range ordered with similar correlation length. This observation suggests entangled charge and lattice instabilities in CsV$_3$Sb$_5$ rather than the intertwined charge and spin correlations that are widely believed to be crucial for the cuprate high-$T_c$ superconductors[1].

Finally, we investigate the unidirectional $1\times4$ superlattice (4a$_O$ phase) that was originally observed by scanning tunneling microscope/spectroscopy on the Sb surface[22]. The 4a$_O$ phase emerges below an intermediate temperature $T^*<60$ K and has been proposed as a possible analogy to the stripe order in cuprates[1] and the cascade ordering in twisted bilayer graphene[33]. Figure 4a shows the topographic image of the clean Sb surface. In agreement with previous studies[22], the Fourier transformation of the topography, shown in Fig. 4b, reveals both surface $2\times2$ and $1\times4$ superlattice modulations. As highlighted by the black dashed lines in Fig. 4a and the red circle in Fig. 4b, both the topographic imaging and the Fourier transformation result confirm the existence of the $1\times4$ superlattice on the Sb surface. While the Fourier transformed $2\times2$ and $1\times4$ superlattice are of comparable intensities on the Sb surface, the 4a$_O$ phase has been

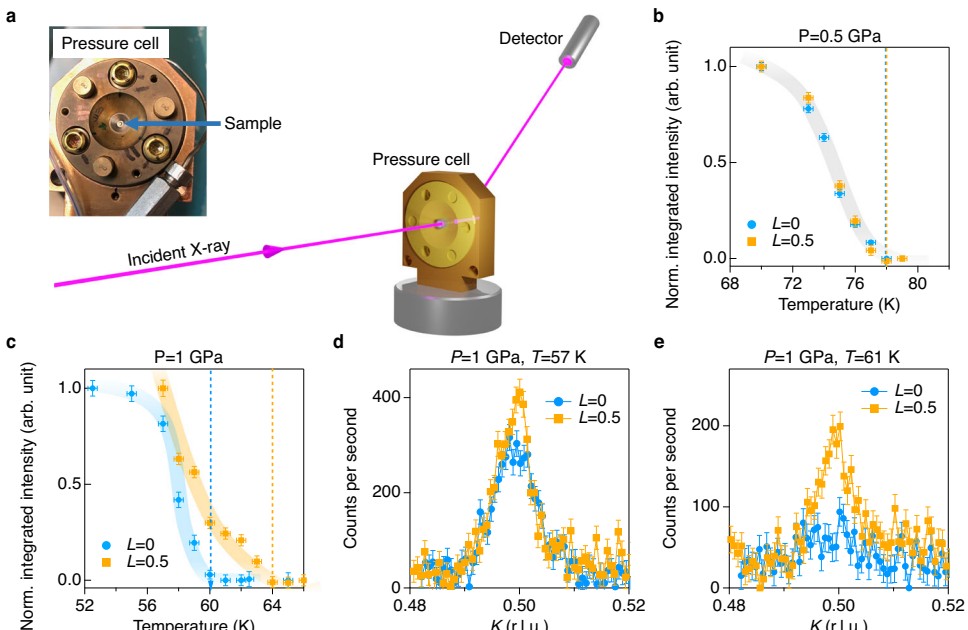

**Fig. 3 | Evolution of conjoined CDWs under pressure. a** The experimental geometry of high-pressure X-ray diffraction. **b, c** Normalized integrated CDW intensity vs temperature taken at $\mathbf{Q}_{CDW}^{2\times2\times2}$=(3, 0.5, 0.5) and $\mathbf{Q}_{CDW}^{2\times2\times1}$=(3, 0.5, 0) at $P$ = 0.5 GPa and 1 GPa. Dashed lines mark the onset temperature of the $2\times2\times2$ CDW (yellow) and $2\times2\times1$ CDW (blue). The intensity in **b** and **c** is normalized by the intensity at the lowest temperature in each curve. The high-pressure X-ray scattering were taken at 20 keV in a transmission geometry. The shadings in **b** and **c** are guides to the eye. **d, e** Direct comparison of the CDW peak intensity of $\mathbf{Q}_{CDW}^{2\times2\times2}$ = (3, 0.5, 0.5)

and $\mathbf{Q}_{CDW}^{2\times2\times1}$ = (3, 0.5, 0) under 1 GPa. The measurements are taken at $T$ = 57 K and 61 K. which is below and above the onset temperature ($T$ = 60 K) of the $2\times2\times1$ CDW (L = 0) under 1 GPa (blue dashed line in **c**). The temperature was determined by the thermal diode reading on the sample stage, which is stable on the level of 0.1 K. We estimated the upper limit of temperature error based on the $T$-dependent order parameter measurement shown in Fig. 1c of the main text, which is less than 0.3 K. The vertical error bars in **b–e** represent 1-standard deviation assuming Poisson counting statistics.

transparent to thermodynamic, Raman and X-ray diffraction measurements[13,18,34,35]. These observations indicate that the STM observed $4a_0$ phase that has a correlation length greater than 30 nm[22] is either a pinning of short-ranged bulk state on the Sb surface or a pure surface effect. To examine the possible short-ranged bulk state, we enhance X-ray sensitivity by (i) enhancing the Sb $5p$ electron contributions at the Sb $L_1$-edge (Fig. 2b); (ii) suppressing fluorescence background intensity below the Sb $L_1$-edge (Fig. 2b); and (iii) suppressing inelastic background intensity using meV-energy-resolution X-ray diffraction (see detailed descriptions in Supplementary Note 4 and 5, and Supplementary Fig. 5 and 6). As we summarize in Fig. 4c–f, the $4a_0$ superlattice peak is absent in all these measurements. Based on experimental sensitivity (see discussions in Supplementary Note 4), we estimate that at 10 K, the diffraction intensity of the $4a_0$ phase, if present, is more than 4-order magnitude smaller than the $2\times2\times2$ CDW that can break $C_6$ to $C_2$[11,12]. We therefore conclude that the $4a_0$ phase, if present in the bulk, is not responsible for electronic nematicity observed in various bulk probes[25,36,37].

In summary, using resonant tender X-ray scattering and high-pressure X-ray scattering, we demonstrate the coexistence of the $2\times2\times1$ CDW in the kagome sublattice and the $5p$-electron assisted $2\times2\times2$ CDW in CsV$_3$Sb$_5$. The observation of conjoined CDWs performatively goes beyond the minimal framework of kagome electronic bands near van Hove filling, and thus provides critical information to resolve the persisting puzzle of CsV$_3$Sb$_5$.

## Methods
### Sample preparation and characterizations
Single crystals of CsV$_3$Sb$_5$ were grown from Cs ingot (purity 99.9%), V powder (purity 99.9%) and Sb grains (purity 99.999%) using the self-flux method[38]. The mixture was put into an alumina crucible and sealed in a quartz ampoule under partial argon atmosphere (~ 0.2 atm at 1273 K). The sealed quartz ampoule was heated to 1273 K at 100 K/h

and kept there for 24 h to ensure the homogeneity of melt. Then it was cooled down to 1173 K at 50 K/h and further to 973 K with 2 K/h. Finally, the ampoule was taken out from the furnace and put in a centrifuge upside down. The ampoule was centrifuged at 50 $g$ for 20 s to separate CsV$_3$Sb$_5$ single crystals from the flux. Except sealing and heat treatment procedures, all other preparation processes were carried out in an argon-filled glove box in order to prevent the reaction of Cs with air and water. The obtained crystals have a typical size of $2\times2\times0.1$ mm$^3$. CDW transition is clearly observed in specific heat measurement as shown in the inset of Fig. 1d.

### Resonant elastic X-ray scattering
Resonant single crystal X-ray diffraction was performed at the 4-ID-D beamline of the Advanced Photon Source (APS), Argonne National Laboratory (ANL). The X-rays higher harmonics were suppressed using a Si mirror and by detuning the Si (111) monochromator. Diffraction was measured using a vertical scattering plane geometry and horizontally polarized ($\sigma$) X-rays. The incident intensity was monitored by a He filled ion chamber, while diffraction was collected using a Si-drift energy dispersive detector with ~200 eV energy resolution. The probed absorption edges are close in energy (4.1–5.5 keV); thus, the use of this detector is key to reject the fluorescence background. The sample temperature was controlled using a He closed cycle cryostat and oriented such that X-rays scattered from the (001) surface.

### High-pressure X-ray diffraction
High-pressure single crystal X-ray diffraction was also measured at the 4-ID-D beamline of the APS, ANL. High pressure was generated using a modified Merrill-Bassett-type diamond anvil cell[39] fitted with a pair of Boehler-Almax anvils of 800 μm culet diameter. A stainless-steel gasket was indented to 70 μm and a sample chamber of 400 μm diameter was laser cut. 4:1 methanol:ethanol was used as pressure media. Diffraction was measured in the transmission geometry in which X-rays

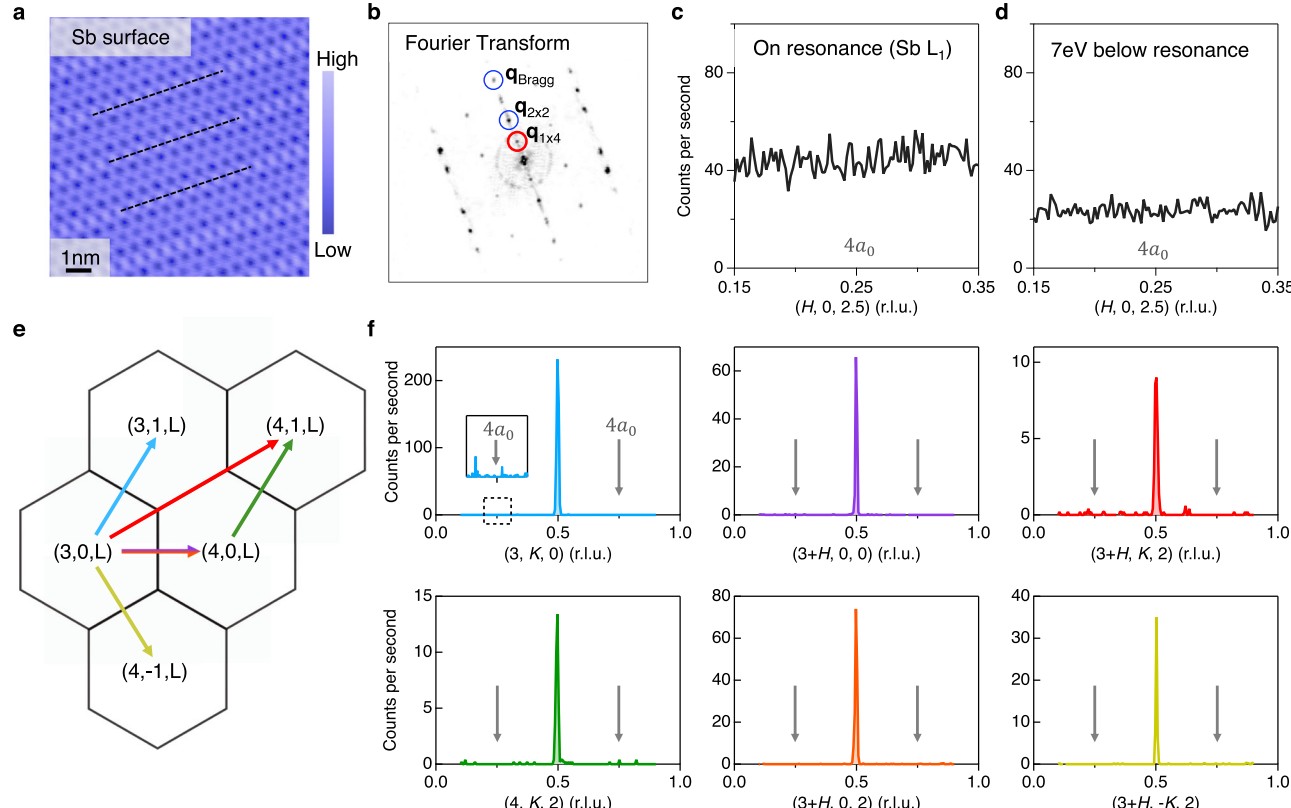

**Fig. 4 | Absence of 1 × 4 superlattice peak in the X-ray diffraction measurement of bulk CsV₃Sb₅. a** Topographic image of a clean Sb surface ($V = -100$ mV, $I = 0.5$ nA). **b** Fourier transform of the topography showing Bragg peaks and charge ordering vector peaks. The in-plane 2 × 2 CDW, 1 × 4 charge order and Bragg peaks are highlighted by circles. **c, d** On-resonance (Sb $L_1$ edge, Fig. 2b) and under-resonance X-ray diffraction measurement around **Q** = (0.25, 0, 2.5) at $T = 10$ K.

**e** Schematics indicating the scattering trajectories for scans shown in **f**. meV-resolution hard X-ray diffraction measurements shown in **f** were performed at 10 K and covered all in-plane high symmetry directions with zero and finite $L$ components. The bulk CDW peaks can be clearly resolved in all scans in **f**, but there is no peak feature at $q = 0.25$ in neither the resonance measurements (**c, d**) nor the high-precision hard X-ray measurement (**f**).

penetrate through both diamond anvils and sample. The sample was cut into ~80 × 80 × 40 μm³ and oriented such that the c-axis is parallel to the X-ray direction when $\theta = 0°$. Temperature was controlled using a He closed cycle cryostat. During the measurement, a piece of Au foil is placed next to the sample in the high-pressure cell. Pressure was calibrated as a function of temperature using the Au lattice constant[40] and controlled in-situ using a He gas membrane. X-ray energy of 20 keV was used to minimize the diamond anvil attenuation. The incident intensity was measured using a $N_2$ filled ion chamber, and diffraction was collected using a NaI scintillator.

**meV-resolution hard X-ray diffraction**
High-precision X-ray scattering measurements shown in Fig. 1c,d and Fig. 4f was taken at 30-ID-C (HERIX), where the highly monochromatic X-ray beam of incident energy $E_i = 23.7$ keV ($\lambda = 0.5226$ A) was focused on the sample with a beam cross section of 35 × 15 μm² (horizontal × vertical). The total energy resolution of the monochromatic X-ray beam and analyzer crystals was $\Delta E$~1.5 meV (full width at half maximum). The measurements were performed in transmission geometry.

**Scanning tunneling microscopy**
Single crystals with size up to 2 mm × 2 mm were cleaved mechanically in situ at 77 K in ultra-high vacuum conditions, and then immediately inserted into the microscope head, already at Helium-4 base temperature (4.2 K). Topographic images in this work were taken with the tunneling junction set up $V = 100$ mV and $I = 0.05$ nA for exploration of areas typically 400 nm × 400 nm. When we found atomically flat and defect-free areas, we took topographic images with the

junction set up $V = 100$ mV and $I = 0.5$ nA to resolve the atomic lattice structure as demonstrated in the paper. Tunneling conductance spectra were obtained with an Ir/Pt tip using standard lock-in amplifier techniques with a lock-in frequency of 997 Hz and a junction set up of $V = 50$ mV and $I = 0.5$ nA, and a root-mean-square oscillation voltage of 0.3 mV. Tunneling conductance maps were obtained with a junction set up of $V = 50$ mV and $I = 0.3$ nA, and a root-mean-square oscillation voltage of 5 mV.

## Data availability
The data that support the findings of this study are available from the corresponding author on reasonable request.

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

## Acknowledgements

This research was sponsored by the U.S. Department of Energy, Office of Science, Basic Energy Sciences, Materials Sciences and Engineering Division (REXS and high-pressure X-ray diffraction and STEM). This research uses resources (REXS and high-pressure X-ray scattering at beam line 4ID and meV-IXS experiment at beam line 30-ID) of the Advanced Photon Source, a U.S. DOE Office of Science User Facility operated for the DOE Office of Science by Argonne National Laboratory under Contract No. DE-AC02-06CH11357 and beamline 4-ID (X-ray scattering) of the National Synchrotron Light Source II, a U.S. Department of Energy (DOE) Office of Science User Facility operated for the DOE Office of Science by Brookhaven National Laboratory under Contract No. DE-SC0012704. Extraordinary facility operations are supported in part by the DOE Office of Science through the National Virtual Biotechnology Laboratory, a consortium of DOE national laboratories focused on the response to COVID-19, with funding provided by the Coronavirus CARES Act. H.C.L. was supported by National Natural Science Foundation of China (Grant No. 12274459), Ministry of Science and Technology of China (Grant No. 2018YFE0202600) and Beijing Natural Science Foundation (Grant No. Z200005) (Sample growth). J.G.C. and J.P.S. was supported by National Natural Science Foundation of China (Grants No. 12025408, No. 11904391) (High-pressure X-ray scattering). Z.Q.W is supported by the U.S. Department of Energy, Basic Energy Sciences Grant No. DE-FG02-99ER45747 and by the Cottrell SEED Award No. 27856 from Research Corporation for Science Advancement (Theory). B.Y. acknowledges the financial support by the European Research Council (ERC Consolidator Grant "NonlinearTopo", No. 815869) (Theory). R.T. acknowledges support from the Deutsche Forschungsgemeinschaft (DFG, German Research Foundation) through QUAST FOR 5249-449872909 (Project P3), through Project-ID 258499086-SFB 1170, and from the Würzburg-Dresden Cluster of Excellence on Complexity and Topology in Quantum Matter–ct.qmat Project-ID 390858490-EXC 2147. Experimental and theoretical work at Princeton University was supported by the Gordon and Betty Moore Foundation (GBMF4547 and GBMF9461; M.Z.H.) and the U.S. DOE under the Basic Energy Sciences program (grant no. DOE/BES DE-FG-02-

05ER46200). J.X.Y. acknowledges support from South University of Science and Technology of China principal research grant (No. Y01202500). We thank Matthew Brahlek, Kun Jiang, Jiaqiang Yan and Raphael Fernandes for stimulating discussions.

## Author contributions

H.M. conceive and designed the research. H.L., G.F., A.H.S., J.P.S., Y.Y.P., C.S.N., J.G.C., H.N.L. and H.M. performed x-ray scattering measurements. Y.X.J, J.X.Y. and M.Z.H. carried out the STM study. S.Y. and A.R.L. performed the STEM measurement. Q.W.Y., C.S.G., Z. J.T. and H.C.L. synthesized the high-quality single crystal samples. H.L., J.X.Y. and H.M. analyzed the experimental data with theoretical input from Z.W., B.Y. and R.T. H.L., J.X.Y. and H.M. prepared the manuscript with inputs from all authors.

## Competing interests

The authors declare no competing interests.
