## [Peer Review File · Nature Communications]

REVIEWER COMMENTS

Reviewer #2 (Remarks to the Author):

Since the previous version, the authors have significantly improved the manuscript by adding new resonant and high-pressure x-ray scattering data, and have found evidence for conjoined charge density waves. These results provide fresh information for understanding the charge density wave and its interplay with the superconductivity, which adds great value to this work. As a result, I think this work can be published in *Nature Communications*, if the following two main points are resolved.

1. It is true that the resonant responses at $L=\text{integer}$ and $L=\text{half integer}$ are different, suggesting conjoined charge density waves. However, it has been argued that in CVS compounds, $2 \times 2 \times 2$ CDW could coexist with $2 \times 2 \times 4$ CDW. Both of these two configurations have contributions at wave-vectors of $2 \times 2 \times 1$ and $2 \times 2 \times 2$, and could, in principle, lead to different responses at $L=\text{integer}$ and $L=\text{half integer}$. How to eliminate this possibility?
2. The observation of different onset temperatures of $2 \times 2 \times 1$ and $2 \times 2 \times 2$ CDWs at $P=1\text{GPa}$ is novel and interesting. To make this conclusion more solid, it would be appreciated to show error bars for temperature in Figure 3 and clarify how the magnitude of peak intensity is normalized in Fig. 3c. This is because in the current plots, it seems that there is still a broad peak at $L=0$ at 61K .

Minor points:

1. First paragraph: "Concomitant electronic nematicity and time-reversal symmetry breaking have been observed." ---Strictly speaking, the onset temperatures for nematicity and TRS in CVS are still open questions, see Ref. 25, arXiv: 2107.10714 and Phys. Rev. Research 4, 023244 (2022). Thus, it would be better to soften the language here.
2. Method: the determination of pressure in the X-ray measurements should be given in the method section. It will allow comparison between different techniques for understanding the P-T phase diagram.

Reviewer #3 (Remarks to the Author):

The modification and clarifying of specific details from the original submission has been made. We thank the authors for their attention to detail in the responses especially the detailed rebuttal submission.

For example "Experimental verification of bulk PDW and the microscopic origin of PDW in cuprates and

CsV₃Sb₅ are still an active research field. In our opinion, while the pair-density modulation in CsV₃Sb₅ phenomenologically resembles the PDW in the cuprates, the underlining mechanisms are likely different. For instance, the cuprates show strong electron-electron correlations and intertwined spin-charge excitations, whereas CsV₃Sb₅ shows weak on-site Coulomb interaction and intermediate electron-phonon couplings. Our new observation of conjoined CDWs and its intimate correlation with double domed superconductivity in CsV₃Sb₅ put extra spatial symmetry constraints on this frontier topic." Is an appropriate response to the reviewer questions posed. This and the effect and strength of the PDW contribution if this has a harmonic this is not clearly defined.

We also see that the FT cutout line is also clearly distinguished.

References to other citations that were missing have been replaced and or added.

The methods have now also been clearly updated to remove the ambiguity.

Overall, the reviewer submits that that clarification of the of the submission greatly clarifies the claims made and allows for the reviewer to be convinced of the new claims.

Response to Reviewer #2

Thank you for your careful review of our manuscript. Below we have copied your report *in black* and describe how we have edited the manuscript in response to your comments *in blue*.

Since the previous version, the authors have significantly improved the manuscript by adding new resonant and high-pressure x-ray scattering data, and have found evidence for conjoined charge density waves. These results provide fresh information for understanding the charge density wave and its interplay with the superconductivity, which adds great value to this work. As a result, I think this work can be published in Nature Communications, if the following two main points are resolved.

Response: We thank Referee #2's support for publishing our manuscript in *Nature Communications*.

1. It is true that the resonant responses at L =integer and L = half integer are different, suggesting conjoined charge density waves. However, it has been argued that in CVS compounds, $2 \times 2 \times 2$ CDW could coexist with $2 \times 2 \times 4$ CDW. Both of these two configurations have contributions at wave-vectors of $2 \times 2 \times 1$ and $2 \times 2 \times 2$, and could, in principle, lead to different responses at L =integer and L = half integer. How to eliminate this possibility?

Response: We thank Review #2 for this insightful comment. The resonant x-ray scattering measurement at ambient pressure and temperature dependent order parameter studies under high-pressure collectively point to conjoined CDWs in CsV_3Sb_5 . The conjoined CDWs could be $2 \times 2 \times 1 + 2 \times 2 \times 2$ as we proposed in our paper or the $2 \times 2 \times 2 + 2 \times 2 \times 4$, as suggested by Reviewer #2. However, existing experimental data showed that the $2 \times 2 \times 4$ CDW varies strongly from sample to sample (*Li et al., PRX 11, 031050; Ortiz et al., PRX 11, 041030; Chen et al., PRL 129, 056401; Stahl et al., PRB 105, 195136; Oey et al., PRM 6, L041801; Xiao et al., arXiv: 2201.05211*). For instance, *Ortiz et al. (PRX 11, 041030)* reported the $2 \times 2 \times 4$ peaks at $T=15$ K, which disappeared at 110 K. *Chen et al. (PRL 129, 056401)* shows the L quarter peak has no change up to 190 K, well beyond the CDW transition. *Stahl et al. (PRB 105, 195136)* reported that the $2 \times 2 \times 4$ superstructure only exists between 60K and 92K (T_{CDW}). Moreover, the K and Rb compounds and lightly Sn doped CsV_3Sb_5 do not host the $2 \times 2 \times 4$ superstructure.

While we consistently observed the selective resonant enhancement at L =half-integer at $T=10$ K in all our samples (5 different pieces), the $2 \times 2 \times 4$ superlattice peaks were only observed in two of our samples, including one sample that shows the $2 \times 2 \times 4$ superlattice peaks even at the room temperature, as shown in Fig. R1 below. We therefore conclude that the $2 \times 2 \times 4$ CDW does not play a key role for the conjoined CDWs. Looking forward, we believe our resonant x-ray scattering study would motivate further studies on the Rb and K compounds as well as chemical substituted CsV_3Sb_5 .

In the revised supplementary materials, section "S6: Out-of-plane structural anomaly revealed by scanning transmission electron microscope (STEM)", we added: "*Besides the L =half-integer CDW peak, superlattice peaks at L =quarter-integer, corresponding to a $2 \times 2 \times 4$ CDW, have been reported in several X-ray studies⁹⁻¹⁴. However, existing experimental data showed that the $2 \times 2 \times 4$*

CDW varies strongly from sample to sample. For instance, Ortiz et al.¹⁰ reported the $2 \times 2 \times 4$ CDW peaks at $T=15$ K, which disappeared at 110 K. Chen et al.¹¹ showed that the L =quarter-integer peaks have no change from 2K to 190K, well beyond the CDW transition. Stahl et al.¹² reported that the $2 \times 2 \times 4$ CDW only exists between 60K and 92K (T_{CDW}). Moreover, the K and Rb compounds and Sn doped CsV_3Sb_5 do not show the $2 \times 2 \times 4$ superstructure¹³. In our study, while the selective resonant enhancement at L =half-integer is consistently observed in all our samples (5 different pieces), the $2 \times 2 \times 4$ superlattice peaks were only observed in two of our samples. These results suggest that the $2 \times 2 \times 4$ CDW does not play a key role for the conjoined CDWs.”

Figure R1. Temperature dependent XRD measurement from $Q=(1, -0.5, 6.05)$ to $(1, -0.5, 6.75)$. The CDW peak at $L=6.5$ disappear above the $T_{CDW} \sim 92$ K, whereas the superstructure peak at $L=6.25$ persist up to room temperature.

2. The observation of different onset temperatures of $2 \times 2 \times 1$ and $2 \times 2 \times 2$ CDWs at $P=1$ GPa is novel and interesting. To make this conclusion more solid, it would be appreciated to show error bars for temperature in Figure 3 and clarify how the magnitude of peak intensity is normalized in Fig. 3c. This is because in the current plots, it seems that there is still a broad peak at $L=0$ at 61K.

Response: We thank Reviewer #2 for this helpful comment. The temperature was determined by the thermal diode reading on the sample stage, which is stable on the level of 0.1 K. We also estimated the upper limit of temperature error based on the T -dependent order parameter measurement shown in Fig. 1c of the main text, which is less than 0.3 K. We added this note in the caption of Fig. 3.

In the revised manuscript, we added temperature error-bar in Figure 3 (See also Fig. R2). The intensity in Figure 3b, c is normalized by the intensity at the lowest temperature in each curve. We added this information in the caption of Fig. 3.

In Fig. 3d, the $L=0$ shows an extremely weak residue, this residual intensity is likely due to the vestigial or fluctuated incoherent state, and it could persist well above the transition. Indeed, the

ambient pressure CDW peak in CVS compound is reported to show residual peak up to ~ 150 K (Phys. Rev. Lett. **129**, 056401).

Figure R2.

Minor points:

1. First paragraph: “Concomitant electronic nematicity and time-reversal symmetry breaking have been observed.” ---Strictly speaking, the onset temperatures for nematicity and TRS in CVS are still open questions, see Ref. 25, arXiv: 2107.10714 and Phys. Rev. Research 4, 023244 (2022). Thus, it would be better to soften the language here.

Response: We thank Reviewer #2 for pointing this out. We rewrite this sentence as “*Electronic nematicity and time-reversal symmetry breaking that are potentially related to the CDW phase have been observed.*”

2. Method: the determination of pressure in the X-ray measurements should be given in the method section. It will allow comparison between different techniques for understanding the P-T phase diagram.

Response: We added this part into the Method “*During the measurement, a piece of Au foil is placed next to the sample in the high-pressure cell. Pressure was calibrated as a function of temperature using the Au lattice constant⁴¹ and controlled in-situ using a He gas membrane.*”